# Progress and Setbacks in Translating a Decade of Ferroptosis Research into Clinical Practice

**DOI:** 10.3390/cells11142134

**Published:** 2022-07-06

**Authors:** Friedrich Alexander von Samson-Himmelstjerna, Benedikt Kolbrink, Theresa Riebeling, Ulrich Kunzendorf, Stefan Krautwald

**Affiliations:** Department of Nephrology and Hypertension, University Hospital Schleswig-Holstein, 24105 Kiel, Germany; friedrich.vonsamson-himmelstjerna@uksh.de (F.A.v.S.-H.); benedikt.kolbrink@uksh.de (B.K.); theresa.riebeling@uksh.de (T.R.); kunzendorf@nephro.uni-kiel.de (U.K.)

**Keywords:** ferroptosis, immunogenic cell death, therapeutic approaches, clinical outcome

## Abstract

Ten years after its initial description, ferroptosis has emerged as the most intensely studied entity among the non-apoptotic forms of regulated cell death. The molecular features of ferroptotic cell death and its functional role have been characterized in vitro and in an ever-growing number of animal studies, demonstrating that it exerts either highly detrimental or, depending on the context, occasionally beneficial effects on the organism. Consequently, two contrary therapeutic approaches are being explored to exploit our detailed understanding of this cell death pathway: the inhibition of ferroptosis to limit organ damage in disorders such as drug-induced toxicity or ischemia-reperfusion injury, and the induction of ferroptosis in cancer cells to ameliorate anti-tumor strategies. However, the path from basic science to clinical utility is rocky. Emphasizing ferroptosis inhibition, we review the success and failures thus far in the translational process from basic research in the laboratory to the treatment of patients.

## 1. Introduction

Ferroptosis is a mode of cell death that has recently garnered immense research attention. Since its first description in 2012, there are now well over 4000 ferroptosis-related publications on PubMed. Ferroptosis was initially discovered when Dixon and colleagues screened small molecular substances to induce death in *RAS*-mutated cancer cells [1]. They reported that the *RAS*-selective lethal (RSL) compounds RSL3 and erastin initiated an iron-dependent type of regulated cell death (RCD) that resulted in the lipid-peroxide-mediated breakdown of cellular membranes. While they were unable to rescue cells using inhibitors of the classic pathways of RCD, such as apoptosis or necroptosis, the lipophilic radical trapping agent ferrostatin-1 (Fer-1) blocked ferroptotic cell death efficiently. Thus, in most ensuing studies a definition of ferroptosis has been used that requires at least one of two criteria to be met: first, cell death that is induced by a specific ferroptosis inducer such as RSL3 or erastin and second, cell death that can be blocked by specific ferroptosis inhibitors such as Fer-1 or liproxstatin-1 (Lip-1). Subsequently, the molecular mechanisms of ferroptosis were investigated in detail and its therapeutic implications were explored. In addition to its potential role in cancer therapy, ferroptosis is an important pathophysiological mechanism contributing to acute kidney injury (AKI), neurological diseases, cardiomyopathy, and liver failure. Therefore, two general concepts for the therapeutic use of ferroptosis modulation seem plausible: to induce ferroptosis in susceptible cancer cells, or to inhibit it where ferroptotic cell death leads to organ dysfunction. In this review, we briefly recapitulate the basic molecular mechanisms of ferroptosis, examine its pathophysiological role in diseases, and summarize the current knowledge on the therapeutic use of ferroptosis modulation.

### 1.1. Molecular Mechanisms of Ferroptosis

The morphological hallmarks of ferroptosis are condensed to be the presence of bilayer membranes, decreased mitochondrial volume [1], and according to some reports, cellular swelling [2,3]. These changes result from nanopores formed as a consequence of high lipid peroxidation levels in the phospholipid pool of the cellular membranes [2,3]. This phospholipid pool is particularly vulnerable to lipoxygenase (LOX)-driven peroxidation when it contains polyunsaturated fatty acyls (PUFAs), as hydrogen abstraction from PUFAs containing bis-allylic protons easily produces peroxyl radicals [4]. Acyl-CoA synthetase long-chain family member 4 (ACSL4) plays a critical role in this regard, as it actively increases the pool of membrane lipids containing PUFAs, dictating ferroptosis sensitivity. In this context, it was demonstrated that ferroptosis could not occur in *Acsl4*-knockout NIH3T3 cells [5]. It can be assumed that ferroptotic cell death can only ensue when plasma membrane PUFA levels are sufficiently high [6] and when free reactive iron is available [1]. Ferroptosis is an iron-dependent process for two reasons: LOX needs iron to sustain enzyme activity [7], reviewed in [8], and iron can directly oxidize phospholipids in a LOX-independent Fenton reaction [9]. Usually, iron is taken up by cells in form of transferrin through the transferrin receptor 1 (TfR1) and then safely stored in ferritin complexes, but the breakdown of these complexes through lysosomal and autophagic mechanisms can release iron, increase the labile iron pool and sensitize for ferroptotic cell death in a process termed ferritinophagy (reviewed in [10]). These reactions consistently happen in all cells but typically do not lead to cell death, as cells are equipped with multiple antioxidative defense systems. The first and arguably most significant of such systems is the glutathione peroxidase 4 (GPX4) system [11]. Therein, the enzyme x_c_^−^ controls the cellular import of cystine in exchange for glutamate at the plasma membrane, which is a necessary step for producing the GPX4 substrate glutathione (GSH), catalyzed by glutamate cysteine ligase (GCL) and GSH synthetase. By oxidizing GSH, GPX4 can directly reduce peroxidized lipids in the plasma membrane, cytosol, and organelles such as the mitochondria [11,12]. Therefore, inhibiting cystine import and direct GPX4 inhibition are two different means of inducing ferroptotic cell death through impaired lipid peroxide detoxification [1,11]. However, plasma membrane phospholipids can also be rescued from lipid peroxidation independently of GPX4 by coenzyme Q_10_ (CoQ_10_); ferroptosis suppressor protein 1 (FSP1) acts as an oxidoreductase of CoQ_10_, which shuttles reducing agents into the lipid bilayer plasma membrane [13,14]. Additionally, mitochondrial membrane peroxidation can be reversed by dihydroorotate dehydrogenase (DHODH), which can act parallel to mitochondrial GPX4 by reducing CoQ_10_ to CoQ_10_H_2_ [12]. Another protective enzyme is GTP cyclohydrolase 1 (GCH1), which is the rate-limiting enzyme for synthesizing tetrahydrobiopterin (BH4) and dihydrobiopterin (BH2). The GCH1–BH4–BH2 system protects PUFA phospholipids with two polyunsaturated acyl chains from ferroptotic degradation independently of GPX4; furthermore, it can also reduce CoQ_10_ [15,16]. In summary, ferroptotic cell death is executed through phospholipid peroxidation that leads to plasma membrane pores, measuring a mere few nanometers, but cells are equipped with multiple systems that reverse lipid peroxidation. For a more detailed description of the molecular mechanisms of ferroptosis, please refer to recent and extensive reviews covering this topic [17,18].

### 1.2. The Purpose of Ferroptosis

Before proceeding with more clinically related aspects, it is important to discuss the innate role of ferroptosis. Clearly, ferroptosis involves highly regulated and programmed molecular mechanisms, but as opposed to apoptosis, necroptosis, and pyroptosis, which are important contributors to ontogeny and pathogen control (reviewed in [19,20,21]), the physiological role of ferroptosis appears to be less clearly defined. As most of the known regulated processes involving ferroptosis are directed at detoxifying reactive oxygen species (ROS) to maintain plasma membrane integrity, it can be concluded that ferroptotic cell death merely occurs as a detrimental byproduct of the oxygen-based, high energy-consumption metabolism in humans, and as such, it is an undesirable consequence of an inevitable problem: ROS constantly accumulate intracellularly during critical cellular functions and at times will overwhelm the cellular antioxidative systems. Situations that generate ROS include ATP production via the respiratory electron chain (reviewed in [22]), CYP-mediated hormone synthesis [23], infection (reviewed in [24]), ionizing radiation [25], and toxic events (reviewed in [26]).

Yet, some of the recent advances in our understanding of ferroptosis challenge this simplistic view: for example, it was discovered that protein kinase C βII (PKCβII) is an important regulator of ferroptosis, as it actively amplifies pro-ferroptotic ACSL4 signaling in the presence of ferroptotic stimuli via the phosphorylation-dependent promotion of ACSL4 dimerization [27]. Hence, the question is, if ferroptosis is an unwanted process, counteracted by an arsenal of antioxidative defense systems, then why are cells equipped with such a mechanism that perpetuates cell death? Interestingly, it was demonstrated that ferroptotic cell death occurs in a synchronized manner, spreading through the kidney tubuli upon ferroptosis induction [28]. Riegman et al. reported similar findings in HAP1 cells [2], where a wave of death spread through the cell population after treatment with ferroptosis-inducing agents. Furthermore, after blocking cell death using osmoprotectants, the authors continued to register wave-like signaling, indicating ferroptotic cellular communication. Why would cells be programmed to continue death signaling to their neighboring cells? Does this not imply an evolutionarily conserved beneficial function of ferroptosis? Providing some answers along these lines, Katikaneni et al. elegantly demonstrated in zebrafish tail fins how long-range lipid peroxidation-mediated signaling was used for leukocyte chemotaxis, contributing to tissue wound closure [29]. Hence, lipid peroxidation appears to be useful in certain situations, as long as it is not excessive, as that would trigger ferroptotic death and potentially aggravate tissue injury.

An intriguing concept explaining how ferroptotic cell death itself can be beneficial to the organism approaches the origins of ferroptosis research more closely: ferroptosis as an anti-cancer strategy. Whereas initial publications from the Stockwell group focused on developing substances to induce ferroptotic cell death in *RAS*-mutated cancer cells [1,30], several clues have surfaced since, demonstrating that ferroptosis is an innate, intrinsically occurring anti-cancer mechanism. Some of the most significant anti-tumor suppressors, namely p53 and BAP1, downregulate SLC7A11 (a subunit of x_c_^−^), which indirectly impairs GPX4 function [31,32]. Additionally, the immune system sensitizes tumor cells to ferroptosis as a means of regulating the body’s response to cancer. In this context, Wang et al. reported that CD8^+^ T cells activated by preceding immunotherapy used interferon-gamma (IFNγ) to downregulate SLC3A2 (another subunit of x_c_^−^) and SLC7A11 in the tumor microenvironment, thereby reducing intracellular cystine levels and priming tumor cells towards ferroptosis [33]. Fittingly, Efimova et al. discovered that early ferroptotic MCA205 cells passively released the damage-associated molecular patterns (DAMPs) ATP and HMGB1, which in turn stimulated nearby dendritic cells to use phagocytosis and reduce tumor load, improving the survival of mice in a cancer model [34].

With the increasing understanding of ferroptosis in health and disease, two potential therapeutic approaches have emerged: the inhibition of ferroptosis in situations where it causes excessive cell death and organ damage, and the induction of ferroptosis as a treatment of certain types of cancer. Ten years after the term ferroptosis was coined, where do we stand? How far has the translation from bench to bedside progressed? The subsequent sections will elaborate on the recent evidence from animal and human studies, emphasizing the current knowledge of ferroptosis inhibition.

## 2. Therapeutic Relevance of Ferroptosis Inhibition

The initial description of ferroptosis sparked a plethora of studies that evaluated its effect on organ damage in animal disease models. The reasoning was straightforward: numerous pathophysiological processes increase ROS and lipid peroxidation levels, leading to ferroptosis in previously well-functioning cells. If anti-ferroptotic intervention can rescue such cells for a sufficiently long time to remove the disease trigger, it will improve the chances of full organ recovery. The general applicability of this therapeutic concept was tested in animal models of acute tubular necrosis in the kidney [28,35,36,37,38,39], in neurological disease [40,41,42,43], in cardiomyopathy [44,45,46,47,48,49], and in liver failure [36,50,51,52], which all demonstrated the alleviation of organ dysfunction after ferroptosis was blocked.

### 2.1. Ferroptosis Inhibitors

Fer-1 and Lip-1 were separately identified as the first small molecules that can inhibit ferroptosis in vitro and in vivo [1,36]. Subsequently, several additional second- and third-generation ferrostatins were developed because of the poor plasma stability of Fer-1 [28,53,54]. Ferrostatins and Lip-1 exhibit high efficacy as radical-trapping antioxidants (RTA), with a particularly high potency in phospholipid bilayer membranes when compared to other antioxidants [55]. It is important to remember that these compounds are required to undergo rigorous testing before they become potentially clinically available for treating humans, and all available evidence of their in vivo utility is based on their use in animal studies (typically mice). The best route of application and their pharmacological properties in humans remain completely uncertain. Therefore, it would be easier to repurpose substances with ferroptosis-inhibiting characteristics that are already in clinical use for other indications. In this context, deferoxamine (DFO) was shown early-on to be a ferroptosis inhibitor through its effective chelation of free iron, albeit its ferroptosis-inhibiting potency is lower than those of Fer-1 or Lip-1 [1]. Other pharmaceutics that have been reported as ferroptosis inhibitors of at least moderate capacity include vitamin C, rosiglitazone, N-acetylcysteine (NAC), vitamin E and vitamin K1, to name just a few [56,57,58,59,60].

### 2.2. Acute Kidney Injury

AKI is a frequent and dangerous disorder that occurs in up to 20% of all hospitalized patients and is associated with a high mortality [61]. It is defined as a rise in serum creatinine from baseline and/or decreased urine production [62]. Several triggers with significant pathophysiological diversity can cause AKI by damaging the glomerular and tubular components of the nephrons. Accounting for ≥85% of all AKI cases, the most frequently affected renal compartment is the tubules. Acute tubular necrosis (ATN) occurs when tubule damage is severe, and is typically caused by renal hypoperfusion or hypoxemia (reviewed in [63]). Due to its clinical significance, the molecular mechanisms of ATN were extensively studied in the past decades. The mechanisms of regulated necrosis pathways were identified as being drivers of ATN, the most impactful of which is ferroptosis [28,35,36,37,38,64]. Typically, the involvement of ferroptosis in the respective AKI models was demonstrated by inducing kidney injury in mice and rescuing organ function with ferroptosis inhibitors, but some molecular markers (ACSL4, lipid peroxidation levels) were additionally established to indicate the presence of ferroptosis in vivo. Several experimental models of ferroptotic AKI were reported, namely renal ischemia–reperfusion injury (IRI) [28], rhabdomyolysis-induced AKI (RIAKI) [35], unilateral ureteral obstruction [37], folic acid (FA)-induced AKI [38], cisplatin-induced AKI [39], and *Gpx4* knockout-induced AKI [36]. Nevertheless, we focus on renal IRI and RIAKI, as these appear to be the most clinically relevant and most convincingly established models. As an aside, the FA model has not been reproducible by us and others [unpublished data; personal communication] and has no clinical correlate. Based on our own data, the cisplatin model is a FAS- and RIPK1-mediated model in which ferroptosis inhibition is not protective and therefore obsolete, as ween in [64,65] and unpublished data.

The first model studied in the context of ferroptotic AKI was renal IRI, an entity that commonly occurs during kidney transplantation, as several hours can pass between organ removal from the donor and the restoration of blood flow in the recipient, strongly increasing the risk for delayed graft function (DGF) [66]. As the organ is usually stored in hypothermic solutions between removal and implantation, this period is termed “cold ischemia time”. Other clinical scenarios in which renal IRI occurs are surgeries, shock, or circulatory arrest [67]. In the corresponding mouse model of renal IRI, ferroptosis inhibitors are typically administered shortly before IRI induction [28], a fact that is important to acknowledge, as such a preventative strategy requires control over the timing of damage induction. While the protective effects of ferroptosis inhibition in renal IRI are reproducible [60,68,69,70], clinically, neither a small trial evaluating vitamin C [71] nor a large randomized controlled trial (RCT) evaluating NAC in human kidney transplantation demonstrated benefit in terms of DGF incidence [72]. Thus far, efforts to prevent IRI in kidney transplantation have focused on mechanical solutions such as bridging cold ischemia time with artificial machine perfusion, successfully reducing DGF risk by ≥50% [73,74]. This approach should be considered an opportunity, as it is possible to use the machine perfusates as vectors for delivering medication to the organs [75]. Therefore, it appears promising to supplement perfusates with ferroptosis inhibitors to further decrease transplant damage.

RIAKI most frequently occurs in patients with dehydration, traumatic or ischemic muscle injury, or drug-induced rhabdomyolysis [76]. The presence of ferroptosis in RIAKI was studied in a mouse experimental model [35], where mice underwent fluid restriction and subsequently received intramuscular glycerol injections to induce rhabdomyolysis and AKI. Fer-1 and the ferroptosis-inhibiting diarylheptanoid curcumin improved renal function when injected before glycerol application, and this effect was partially preserved when curcumin was administered therapeutically after the induction of rhabdomyolysis. However, renal protection in the therapeutic intervention group (curcumin administered 3 h post-glycerol) was markedly lower than that in the preventative group (curcumin administered 24 h pre-glycerol). Unfortunately, this limits the clinical utility of these findings, as patients most commonly have experienced the induction of rhabdomyolysis before initial presentation, sometimes several days in advance. To the best of our knowledge, no efforts have been undertaken to evaluate the efficacy of ferroptosis inhibition in human RIAKI, and based on the experimental findings, we would not expect strong protective effects in a real-life setting.

In conclusion, while there are certainly definite clues indicating the pathophysiological role of ferroptosis in AKI (or more specifically, ATN), the evidence for ferroptosis inhibition as a therapeutic strategy in kidney disease is relatively weak. In our opinion, reducing the cold ischemia time-induced damage of kidney transplants with ferroptosis inhibitors combined with machine perfusion appears to be the most promising approach for translation to the clinic.

### 2.3. Neurological Diseases

Diseases of the brain and the nerves occur in widely varying forms and fashions. Such diseases remain the globally leading cause of disability-adjusted life years and the second leading cause of death [77]. The involvement of ferroptosis has been evaluated in the pathophysiology of ischemic and hemorrhagic stroke [40,42], and in the neurodegenerative disorders Parkinson’s disease (PD) [41], Alzheimer’s disease (AD) [43], Huntington’s disease [53], and Friedreich’s ataxia [78]. Here, we focus on inhibiting ferroptosis as a therapeutic concept in stroke, PD, and AD because of the overwhelmingly high prevalence of these entities.

Stroke occurs as a consequence of brain tissue hypoperfusion, which can either be caused by the clotting of the brain arteries leading to ischemia or by intracranial bleeding (ICB) (reviewed in [79]). An experimental mouse study provided evidence that the therapeutic inhibition of ferroptosis can alleviate ischemic brain damage, where IRI was induced by middle cerebral artery occlusion (MCAO) followed by reperfusion. In mice, tau protein caused pro-ferroptotic iron accumulation during stroke, and *Tau*-knockout mice and wild-type mice treated with Lip-1 at reperfusion or 6 h post-reperfusion exhibited significantly better neurological outcomes than vehicle-treated mice, albeit the protective effect of Lip-1 was smaller in the 6 h post-reperfusion group [40]. In another study, the contribution of ferroptosis to brain damage in ICB was reported, where Fer-1 reduced hemoglobin-induced neuronal ferroptosis in vitro and oxidative stress markers were elevated in vivo. Unfortunately, the study did not report data on the in vivo efficacy of Fer-1 in ICB [42]. To the best of our knowledge, there are currently no ongoing trials evaluating ferrostatins or Lip-1 for treating stroke in humans. However, a vast amount of trials from the 1990s and early 2000s evaluated antioxidants such as vitamin C and E for treating ischemic stroke. When these studies were conducted, the concept of ferroptosis had not been described, but retrospectively, they can be considered to be early trials of ferroptosis inhibitors. While many of the pre-clinical and phase I trials provided a reason to perceive a role for antioxidants in stroke treatment, these hopes could not be sustained after a series of negative phase III trials (reviewed in [80]). Instead, thrombolytic therapy and thrombectomy emerged as the guideline-recommended therapies for ischemic stroke (reviewed in [79]). However, substances with ferroptosis-inhibiting characteristics are still being evaluated in clinical studies, and an RCT using NAC in stroke is currently recruiting patients (NCT04918719). Going forward, the combination of antioxidants with thrombolysis or thrombectomy might yield a potentially successful approach.

PD is a neurodegenerative disorder caused by the progressive loss of neurons in the substantia nigra pars compacta of the basal ganglia. It leads to motoric, vegetative, and memory dysfunction with varying degrees of severity and affects 2–3% of the population aged ≥65 years (reviewed in [81]). Lipid peroxidation occurred early in PD [82], and other features of ferroptosis such as GSH depletion and high iron levels were present in the brains of patients with PD [83,84]. Recently, it was reported that Fer-1 alleviated PD symptoms in a mouse model [41]. In the same study, with an admittedly small sample size (*n* = 3), a strong transcriptional upregulation of the anti-ferroptotic enzyme GPX4 was observed in the substantia nigra of human brains. Accordingly, clinical trials have evaluated the therapeutic potential of several antioxidants. However, the placebo-controlled DATATOP trial was unable to demonstrate clinical improvement after vitamin E treatment in a cohort of 800 PD patients [85]. Somewhat encouragingly, some small recent RCTs indicated the alleviation of PD symptoms with NAC or iron-chelating treatment [86,87], while another RCT evaluating iron chelators reported only a trend towards improved clinical outcomes without reaching statistical significance (notably, this study was likely underpowered) [88]. Whether the generally encouraging effects of iron chelators and NAC can be sustained in larger trials remains unknown, but international guidelines currently do not recommend antioxidants or ferroptosis inhibitors for treating PD [89,90] and none of the recently developed ferroptosis inhibitors are presently being tested in clinical trials in this indication.

AD affects approximately 5% of the general population and more than a third of all people aged ≥85 years [91]. It typically causes short-term memory loss, but can also impair expressive speech and other executive mental functions, drastically decreasing the quality of life for patients and relatives alike. Tragically, no effective treatment is available for this devastating condition. Pathophysiologically, the accumulation of amyloid-β-containing extracellular plaques and tau-containing neurofibrillary tangles deranges synaptic function in the medial temporal lobe and later on in other cortical areas of the brain (reviewed in [92]). Hinting at a role of ferroptosis in AD, GPX4 depletion in an inducible brain-specific *Gpx4*-knockout mouse model led to neurodegeneration and neuronal lipid peroxidation, which was exacerbated by vitamin E restriction, but which could be rescued by Lip-1 [43]. Accordingly, amyloid-β-induced neurodegeneration in an AD mouse model was reduced in the presence of Fer-1 and Lip-1, albeit the effects were mild and not completely consistent between the two ferroptosis inhibitors [93]. In a large prospectively evaluated cohort from the Netherlands, high regular intake of vitamin C and E reduced the risk for developing AD [94]. Several controlled trials investigated the potential of vitamin C and E in AD treatment, but meta-analyses demonstrated no improvement of relevant clinical outcomes. On the contrary, high-dose vitamin E might even pose a safety hazard by increasing the occurrence of cardiovascular events [95,96].

In summary, antioxidant treatment has been extensively studied in the context of neurological diseases. Regretfully, despite of glimmers of hope from smaller clinical trials, the outcomes of the larger blinded controlled trials have been disappointing thus far. The recent evidence regarding Fer-1 and Lip-1 efficacy in mouse models brings fresh hope for the use of ferroptosis inhibition as a therapeutic option in neurology. However, keeping in mind the sobering results from vitamin C and E trials, and as both Fer-1 and Lip-1 do not easily cross the blood–brain barrier and must be administered intranasally or even directly into the brain, our expectations for their future success in the treatment of these entities are cautious [40,41].

### 2.4. Heart Disease

Heart disease includes a wide range of disorders that acutely or chronically impair the heart or its blood vessels. Despite slowly decreasing age-standardized death rates, they continue to represent the globally leading cause of death and morbidity [97,98]. The most common entity of heart disease is ischemia caused by atherosclerosis and coronary artery disease, but other processes such as valvular dysfunction, hypertensive hypertrophy, and cardiomyopathy also contribute to the disease burden [99]. Ferroptosis has received attention as a pathophysiological factor in myocardial infarction (MI) [44,45], hypertensive heart failure [46,47], as well as sepsis- [48] and chemotherapy-induced cardiomyopathy [49].

Ischemic heart disease can develop chronically, but an acute occlusion of atherosclerotic coronary arteries often occurs in the disease course, leading to potentially lethal MI (reviewed in [100]). In a mouse model of MI, differential regulation of transcription and protein expression of GPX4 and several other key ferroptotic molecules were observed in heart tissue via RNA sequencing (RNA-seq) and mass spectrometry [44]. Additionally, Lip-1 reduced the infarct size in a mouse model of cardiac IRI, where mouse hearts were excised and perfused with buffer solution [45]. To what extent this truly represents the corresponding disease model in humans is unclear, but this evidence was supported by another animal study in which rats with or without diabetes underwent left anterior descending coronary artery ligation, followed by reperfusion [101]. In diabetic rats undergoing myocardial IRI, ACSL4 upregulation and GPX4 downregulation were reported in the diabetic and IRI groups, and the administration of Fer-1 preserved hemodynamic stability. Unfortunately, the study did not include a non-diabetic IRI + Fer-1 group. Regarding evidence from humans, Bulluck et al. used magnetic resonance imaging to visualize high residual levels of iron in cardiac tissue several days after MI, which correlated with the extent of adverse cardiac remodeling [102]. Somewhat analogously to the studies on antioxidants in stroke, numerous RCTs tested the efficacy of NAC and vitamins in MI. Addressing each RCT is outside the scope of this review, but while many RCTs demonstrated the improvement of surrogate markers such as troponin levels or infarct size or even trends toward better clinical outcomes, no large RCT to date has convincingly demonstrated improved mortality or long-term cardiac function after antioxidant treatment of MI (reviewed in [103,104]). Currently, international guidelines do not recommend antioxidants in the context of MI [105], and RCTs are not evaluating new anti-ferroptotic agents such as Fer-1 or Lip-1.

Heart failure caused by hypertensive heart disease is a highly relevant clinical problem with a prevalence of >200 per 100,000 people [106]. A few pre-clinical studies have indicated ferroptotic involvement in this entity: knockout of the pro-ferroptotic NADPH oxidase 4 (*Nox4*) decreased transverse aortic constriction (TAC)-induced left ventricular remodeling in a mouse model simulating hypertensive heart disease [47], and the pharmacological inhibition of ferroptosis through puerarin prevented TAC-induced reduction of the left ventricular ejection fraction [46]. However, we are not aware of any reports of the more established ferroptosis inhibitors alleviating TAC-induced heart failure or any efforts to implement anti-ferroptotic strategies for treating hypertensive heart disease.

Although currently there is no universally accepted definition of sepsis-induced myocardial dysfunction (SIMD), it is a frequently seen entity in intensive care units (ICUs) and an often lethal complication of sepsis (reviewed in [107]). The state-of-the-art treatment approach to sepsis includes antibiotics, fluid management, and vasopressors or inotropes when needed, but no therapeutic option is specifically directed at preventing SIMD. Therefore, a recent report implicating ferroptosis in lipopolysaccharide (LPS)-induced cardiomyopathy in mice is of interest in this regard [48]. The authors demonstrated that LPS induced ferritin breakdown and consequently increased cellular labile iron pools. While the in vitro data only appointed a very mild protective effect to Fer-1 in the LPS-induced death of H9c2 myofibroblasts, the in vivo data demonstrated a significant improvement in the cardiac lipid peroxidation levels, cardiac function, and survival of mice treated with Fer-1 and the iron chelator dexrazoxane. Interestingly, a recent meta-analysis indicated a possible improvement of outcomes in a subgroup of sepsis patients treated with vitamin C for 3–4 days [108]. It would be interesting to examine the effect of vitamin C in septic patients with SIMD, but such a subgroup analysis was not conducted. In summary, it is too early to recommend ferroptosis inhibition in the context of SIMD. However, future research is warranted.

A final cardiac disorder that should be discussed in the context of ferroptosis, doxorubicin-induced cardiomyopathy (DIC), has attracted some interest. DIC was long considered a cardiotoxic apoptosis model, but several research groups have recently claimed to have found the involvement of other types of RCD such as necroptosis, pyroptosis, and autophagy (reviewed in [109]). Evaluating the role of ferroptosis, Fang et al. reported that doxorubicin treatment led to mitochondrial iron overload and lipid peroxidation. In their mouse model of DIC, Fer-1 not only led to increased survival and reduced cardiac remodeling, but also did so more effectively than the RIPK1 inhibitor necrostatin-1 and the caspase inhibitor emricasan, while mice with RIPK3 deficiency were equally protected [49]. As combining Fer-1 with *Ripk3* knockout conferred the strongest survival benefits, a translational approach could be to strive for the simultaneous inhibition of ferroptosis and RIPK3 before the application of doxorubicin. However, as no RIPK3 inhibitor is currently available in humans, trials evaluating this concept would have to cross large hurdles.

To conclude this section, ferroptosis inhibition has not had its breakthrough in this field. Moreover, raising serum iron levels well above the normal range by regularly infusing a high dose of intravenous ferric carboxymaltose notably improves the quality of life, morbidity and re-hospitalization rate of patients with heart failure and iron deficiency [110,111]. Certainly, high iron levels alone do not suffice to induce ferroptosis, but still, this argues against a strong involvement of ferroptosis in cardiac disease, as one would expect clinical outcomes to worsen if this iron-dependent form of cell death was significantly involved in disease pathophysiology. However, we are certain that further trials evaluating antioxidant treatment in the abovementioned and in yet-to-be-identified cardiac ferroptosis models will be performed, and we await these results with great interest.

### 2.5. Liver Disease

Liver disease can present acutely, e.g., as acute liver failure caused by hepatitis, or chronically, e.g., as liver cirrhosis caused by steatosis, and results in over two million deaths annually worldwide [112]. Considering the central role of iron in the pathophysiology of ferroptosis and the high iron content in the liver (reviewed in [113]), it is unsurprising that ferroptotic cell death contributes to several types of hepatic injury reviewed in [114].

A principal function of the liver is metabolite detoxification. Hepatic capacities can be overwhelmed when toxin concentrations are too high, as is the case with acetaminophen hepatotoxicity, which represents the best studied cause of drug-induced liver injury (reviewed in [115]). As hepatocytes break down acetaminophen with GSH, cellular GSH deposits can be depleted when acetaminophen overdose occurs reviewed in [116]. The analogy to in vitro models of ferroptosis using erastin to deplete GSH is quite obvious, and in a well-designed in vitro study, Lőrincz et al. demonstrated that Fer-1 prevented GSH depletion and loss of viability of primary murine hepatocytes exposed to acetaminophen toxicity [50], which was confirmed by Yamada et al. in a corresponding mouse model [51]. In fact, acetaminophen-induced liver injury represents the prime example of effective clinical ferroptosis inhibition: a mountain of evidence supports the use of NAC for its treatment in humans, and international guidelines strongly recommend using NAC in acute acetaminophen intoxication [117].

Another entity of hepatic disease in which principles of ferroptosis inhibition were shown to provide therapeutic benefit is hemochromatosis (HH). Occurring with a prevalence of approximately 0.1% in Caucasian males, its most common form is caused genetically by a homozygous C282Y mutation in the *HFE* gene which leads to reduction of hepcidin expression and consequentially increased enteric iron uptake. Liver fibrosis is the hallmark clinical manifestation, but other organs such as the skin, pancreas, heart and joints are also involved (reviewed in [118]). Recently, in vivo studies provided evidence that ferroptosis contributes to the pathogenesis of hemochromatosis, as Fer-1 and DFO blocked the ferroptotic death of hepatocytes, and the application of Fer-1 in mouse models of hemochromatosis protected liver function [119]. Whereas Fer-1 has not been tested in humans, iron chelators such as DFO effectively reduce iron load in HH and are guideline-recommended when therapy with serial phlebotomy is not tolerated [120,121].

Similar to the tubules in the kidney, IRI damages hepatocytes and can lead to acute liver injury, posing a threat to organ function in liver transplantation, circulatory arrest, and large surgeries [122]. In this context, Lip-1 significantly reduced IRI-induced hepatic necrosis [36], and other antioxidant approaches showed a benefit in experimental hepatic IRI (reviewed in [123]). However, clinical trials evaluating the efficacy of NAC yielded inconsistent results (reviewed in [124]). Whether ferroptosis inhibition will eventually make its way into the field of liver transplantation remains to be seen; presently, research efforts appear to be focused on machine perfusion strategies (reviewed in [125]) and on applying antioxidant nanoparticles with preferential liver uptake to reduce hepatic IRI [126], and we are not aware of any studies examining these substances in the context of ferroptosis.

One of the most relevant problems in liver care is the transition of healthy hepatic tissue to fibrotic or cirrhotic tissue. Morbidity, mortality, and liver cancer risk are significant when cirrhosis is present (reviewed in [127]). Several underlying entities can cause such remodeling, but most often it is triggered by alcohol, fatty liver disease, or viral hepatitis. Patients with liver cirrhosis have high hepatic iron levels [128]. Considering the importance of iron in ferroptosis, the protective influence of iron-binding transferrin on liver health was evaluated in mice [52]. As mice genetically deficient in transferrin exhibited a transcriptional upregulation of the anti-ferroptotic GPX4 system and as an iron-enriched diet induced liver injury that was blocked by Fer-1 treatment, it was concluded that ferroptosis might contribute to the occurrence of chronic liver disease and that ferroptosis inhibition could potentially delay this process.

As noted above, liver cancer often develops in a cirrhotic liver. Typically, it is detected via ultrasound examination of the liver, elevated tumor markers, or when patients begin exhibiting symptoms, but the clinician might overlook the very early stages of liver cancer with tumors below the detection threshold of ultrasound and laboratory testing. This must be considered when discussing ferroptosis inhibition as a treatment approach to liver cirrhosis, as conversely, ferroptosis induction is an innate anti-cancer strategy and inhibiting this tumor suppression mechanism could potentially harm the patient.

## 3. Therapeutic Induction of Ferroptosis in Cancer

Given the background of its discovery, that is, as a pathway capable of inducing cell death in *RAS*-mutated tumor cells [1], it is unsurprising that ferroptosis has been thoroughly investigated as a therapeutic option in cancer. Due to their fast replication, tumor cells depend on a high energy-consumption metabolism that is accompanied by ROS production, rendering them an intriguing target for ferroptosis induction (reviewed in [129]). Lei et al. recently published a review of the mechanistic details of ferroptosis induction in tumors [130]; therefore, we will only summarize them here briefly. Their review introduced three major strategies of ferroptosis evasion in tumor cells: (i) the reduction of PUFA phospholipid synthesis and peroxidation in the lipid membranes, (ii) limiting the cellular labile iron pool, and (iii) the induction of anti-ferroptotic defense systems. At the same time, these evasive cancer cells feature an Achilles heel with several vulnerabilities: an imbalance of anti-ferroptotic systems, cell type- and state-specific metabolic features with increased PUFA synthesis, and genetic susceptibilities such as loss-of-function mutations in anti-ferroptotic pathways. These weak spots can be exploited to either re-sensitize tumor cells to ferroptosis or to induce ferroptosis as an adjunct to existing therapeutic approaches [130].

Several tumor entities, cell types, and animal models have been evaluated in the context of ferroptosis induction in cancer therapy. There is a vast body of in vitro and experimental in vivo evidence supporting the notion that such a therapeutic approach offers new possibilities for killing cancer cells [131,132,133,134,135,136,137,138]. For example, the expression of zinc finger E-box binding homeobox 1 (ZEB1), a lipogenic regulator promoting the inclusion of long-chain PUFAs into the cell membrane, was positively correlated with GPX4 sensitivity across 610 cancer cell lines [132]. The same study demonstrated that GPX4 inhibition could then reduce the viability of a human-derived, erlotinib-insensitive prostate cancer organoid with high ZEB1 expression and that GPX4 deficiency halted the growth of a melanoma xenograft in mice. Similarly, previously chemotherapy-insensitive xenograft melanomas in mice exhibited sensitivity to the chemotherapeutic agents dabrafenib and trametinib after *Gpx4* was knocked out [133]. Wu et al. described non-cell autonomous ferroptosis inhibition by E-cadherin-dependent signaling as an important mechanism promoting metastasis and demonstrated that antagonism of this mechanism induced ferroptosis sensitivity in a murine orthotopic mesothelioma model [131]. In small cell lung cancer (SCLC), it was possible to identify the ferroptosis-susceptible SCLC subtypes by systematically analyzing programmed cell death pathways, and ferroptosis induction in the corresponding mouse model improved overall survival in these subtypes [134].

There are many more excellent studies on how ferroptosis induction could be useful in treating cancer, ranging from melanoma [132,133], prostate cancer [132], and lung cancer [14,134] to renal cell carcinoma [135], liver cancer [135,137], endometrial cancer [135], breast cancer [136], fibrosarcoma [12], and lymphoma [138]. This strong evidence has led to hopes that the success from animal studies can be translated into treatment for humans, and the ferroptosis inducers sulfasalazine (NCT04205357) and sorafenib (NCT02559778; NCT03247088) are currently being tested as add-on therapeutics in phase I and II trials. A major remaining difficulty is that there is no evidence that the most commonly used experimental ferroptosis inducers, such as erastin and RSL3, can also exhibit their function in humans, a difficulty that is enhanced by the lack of a truly specific ligand–receptor system for inducing ferroptosis. The use of erastin and RSL3 in murine models was possible after adjustments to their molecular structure [11], but their bioavailability and metabolic stability in humans are unclear. Therefore, identifying Food and Drug Administration (FDA)-approved substances with strong ferroptosis-inducing properties continues to be of interest in this regard.

## 4. Conclusions

Lipid peroxidation impairment of cellular viability has been researched for decades, but the recent works characterizing ferroptosis as an oxidative, iron-dependent form of RCD with very specific molecular players and regulators have deepened our understanding of these processes considerably. As a consequence of the expanding volume of basic research, two therapeutic approaches have emerged to translate these findings into impactful medical contributions: the inhibition of ferroptosis to prevent oxidative organ damage, and the induction of ferroptosis in cancer therapy. While evidence from large RCTs is available to support the clinical use of ferroptosis inhibition in a few scenarios, most studies have yielded sobering results (Table 1). At the same time, while the cumulative evidence presenting ferroptosis induction as a novel anti-cancer strategy is quite convincing, clinical outcome data are still lacking. It is now necessary to define, more specifically, the disease states in which ferroptosis inhibition is expected to be of benefit, and to identify more potent, clinically available ferroptosis inhibitors. Whereas the definition of ferroptosis is relatively straightforward in vitro, one of the currently biggest issues is the difficulty of precisely defining the presence of ferroptosis in vivo, particularly in humans (Table 2). In this regard, the development of methods for detecting ongoing ferroptosis are of interest, as demonstrated in a recent work depicting anti-TfR1 antibodies as specific ferroptosis markers [139]. The next decade in ferroptosis research will be decisive in determining whether the grand efforts of the cell death community will translate into a meaningful clinical impact for patients.

## Figures and Tables

**Table 1 cells-11-02134-t001:** **Clinically relevant ferroptosis models and the state of translation into therapeutic application.** Experimental ferroptotic disease models and the respective inhibitors from preclinical trials (Experimental ferroptosis modulators) are presented along with the respective clinical correlates and their established treatments. The far-right columns indicate whether any type of ferroptosis modulator has been tested in the respective setting and whether a ferroptosis inhibitor is recommended in clinical routine. Bone marrow (BM), cardiomyopathy (CM), deferiprone (DFP), deferoxamine (DFO), doxorubicin (DOX), ferrostatin (Fer), glutathione peroxidase 4 (GPX4), ischemia-reperfusion injury (IRI), left anterior descending coronary artery (LAD), liproxstatin-1 (Lip-1), lipopolysaccharide (LPS), 1-methyl-4-phenyl-1,2,3,6-tetrahydropyridine (MPTP), N-acetylcysteine (NAC), percutaneous coronary intervention (PCI), *RAS*-selective lethal compound 3 (RSL3), renin-angiotensin-aldosterone system (RAAS), rhabdomyolysis-induced acute kidney injury (RIAKI), and vitamin (Vit.).

	Experimental Ferroptosis Model	Experimental Ferroptosis Modulators	Clinical Correlate	Clinically Established Therapy	Ferroptosis Modulators Tested in Humans	Ferroptosis Modulation Currently in Routine Clinical Use
**Kidney**	Renal IRI	Fer-1 [28], Fer 16-86 [28], Vit. K1 [60], Vit. C [68], NAC [69], DFO [70]	Kidney Transplantation Circulatory arrest Hemorrhagic shock	Machine perfusion Hemodynamic stabilization Blood transfusion	Vit. C [71], NAC [72]	No
	RIAKI	Fer-1 [35], Curcumin [35]	Crush syndrome	Fluid resuscitation, bicarbonate	---	
**Brain**	Middle cerebral artery occlusion	Lip-1 [40], Fer-1 [42]	Ischemic stroke	Thrombolysis, thrombectomy	Vit. C [80], Vit. E [80], NAC [NCT04918719]	No
	MPTP-induced neurotoxicity	Fer-1 [41]	Parkinson‘s disease	Dopamine restoration, brain pacemaker	Vit. E [85], NAC [86], DFP [87,88]	
	Brain specific GPX4-KO, amyloid-β-induced neurodegeneration	Lip-1 [43,93], Vit. E [43], Fer-1 [93]	Alzheimer‘s disease	Acetylcholinesterase inhibitors, memantine	Vit. C [94], Vit. E [94,95,96]	
**Heart**	Ligation of LAD, ex vivo non-perfusion	Lip-1 [45], Fer-1 [101]	Myocardial infarction	Platelet inhibition, thrombolysis, PCI, surgery	Vit. C [103], Vit. E [103], NAC [104]	No
	Transverse aortic constriction	Puerarin [46]	Hypertensive heart failure	RAAS blockade, diuretics, β-blockers	---	
	LPS-induced CM	Fer-1 [48], DXZ [48]	Sepsis-induced CM	Antibiotics, hemodynamic stabilization	---	
	DOX-induced CM	Fer-1 [49]	Chemotherapy-induced CM	Supportive, switching chemotherapeutics	---	
**Liver**	Acetaminophen hepatotoxicity	Fer-1 [50,51]	Acetaminophen hepatotoxicity	NAC	NAC [117]	Yes
	Diet-induced iron overload, genetic iron overload	Fer-1 [124]	Hemochromatosis	Serial phlebotomy, iron chelation	Iron chelators [125,126]	
	Hepatic IRI	Lip-1 [36], NAC [123]	Liver Transplantation Circulatory arrest Hemorrhagic shock	Hemodynamic stabilization Blood transfusion	NAC [124] --- ---	No
**Cancer**	Xenografts	Piperazine erastin [11], RSL3 [11]	Brain cancer Leukemia	(Radio-)Chemotherapy, surgery Chemotherapy, BM transplantation, stem cell transplantation	Sulfasalazine [NCT04205357], Sorafenib [NCT02559778, NCT03247088]	No

**Table 2 cells-11-02134-t002:** **Difficulties in translating the in vitro definition of ferroptosis into clinical utility.** The left column represents criteria commonly used to detect ferroptosis in vitro or in experimental models. The right column depicts the weaknesses of these criteria in accurately defining ongoing ferroptosis in human disease. Acyl-CoA synthetase long-chain family member 4 (ACSL4), ferrostatin-1 (Fer-1), glutathione peroxidase 4 (GPX4), liproxstatin-1 (Lip-1), N-acetylcystein (NAC), radical trapping antioxidant (RTA), *RAS*-selective lethal compound 3 (RSL3), transferrin receptor 1 (TfR1).

Criteria Defining Ferroptosis in Experimental Models	Limitation of Applying Ferroptosis Criteria in Human Disease
Specific inhibition of cell death through the lipophilic RTAs Fer-1 and Lip-1	- Unknown pharmacodynamics and pharmacokinetics in humans - Not approved for use in humans
Inhibition of cell death through other RTAs (e.g., NAC, curcumin)	- These drugs were originally described in non-ferroptotic settings and have functions apart from inhibition of ferroptosis → Ferroptosis-specific effects are uncertain - Low potency in ferroptosis inhibition compared to Fer-1 and Lip-1
Specific induction of ferroptosis through ferroptosis inducers in vitro (e.g., RSL3, erastin)	- There is no direct clinical correlate to the ferroptosis inducers RSL3 or erastin - There is no specific ligand-receptor system for targeting ferroptosis in vivo - An in vivo model or a clinical entity, in which ferroptosis occurs as the sole pathological process, has not been identified
Differential regulation of biomarkers (e.g., ACSL4, GPX4, TfR1, lipid peroxides)	- Obtaining material for evaluation requires invasive sampling - Sensitivity is probably high, but differential regulation of these markers also occurs in other processes than ferroptosis → Little diagnostic significance (low specificity) - No standard/reference range to compare the level of biomarker expression to

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
