# Peer review of "Progress and Setbacks in Translating a Decade of Ferroptosis Research into Clinical Practice"

_cells, 2022, doi:10.3390/cells11142134_

Round 1

Reviewer 1 Report

suggestions:  Mention the two  therapy applications in the abstract;  revise the title on Purpose of Ferroptosis  Perhaps "functions" rather than a teleological implication; discuss role of Ferritin, Serum iron and transferrin and non transferrin iron;  Provide a diagram of ferrostatin and its interactions in health and disease...

Reviewer 2 Report

The review on the impact of ferroptosis research on clinical practice is well-written and interesting. It presents an updated clear summary of the basic research of ferroptosis and focuses on the various attempts the use the knowledge in vivo in animal models and patients to inhibit ferroptosis in acute kidney injury, neurological diseases and heart and liver diseases, and to promote ferroptosis in cancer. The general picture is that the clinical data are very preliminary.

- I think that a clear definition for the recognition of ferroptosis would improve the work. In particular, it seems that the authors correctly identified ferroptosis when it was rescued by the specific inhibitors Fer-1 and Lip-1. In this contest, a discussion on the clinical use of the two inhibitors would be useful: method of administration, pharmacological properties or other.

- On the other side, I found that the manuscript gives little attention to the role of iron (which names this form of cell death). In particular, iron-chelators are ferroptosis inhibitors and they have been used for a long in models of iron and liver diseases even before the definition of ferroptosis. I think that this should be mentioned in the work. 

Round 2

Reviewer 1 Report

Thanks  Much more clear